# Improvements in Hearing and in Quality of Life after Sequential Bilateral Cochlear Implantation in a Consecutive Sample of Adult Patients with Severe-to-Profound Hearing Loss

**DOI:** 10.3390/jcm10112394

**Published:** 2021-05-28

**Authors:** Ville Sivonen, Saku T. Sinkkonen, Tytti Willberg, Satu Lamminmäki, Hilkka Jääskelä-Saari, Antti A. Aarnisalo, Aarno Dietz

**Affiliations:** 1Head and Neck Center, Department of Otorhinolaryngology—Head and Neck Surgery, Helsinki University Hospital and University of Helsinki, 00029 Helsinki, Finland; saku.sinkkonen@hus.fi (S.T.S.); satu.lamminmaki@hus.fi (S.L.); hilkka.jaaskela-saari@hus.fi (H.J.-S.); antti.aarnisalo@hus.fi (A.A.A.); 2Institute of Clinical Medicine, University of Eastern Finland, 70211 Kuopio, Finland; tytti.willberg@tyks.fi; 3Department of Otorhinolaryngology, Turku University Hospital, 20521 Turku, Finland; 4Department of Otorhinolaryngology, Kuopio University Hospital, 70029 Kuopio, Finland; aarno.dietz@kuh.fi

**Keywords:** cochlear implant, bilateral, sequential, consecutive sample, adult patients, speech recognition in noise, subjective improvement, etiology of hearing loss, correlation

## Abstract

Bilateral cochlear implantation is increasing worldwide. In adults, bilateral cochlear implants (BICI) are often performed sequentially with a time delay between the first (CI1) and the second (CI2) implant. The benefits of BICI have been reported for well over a decade. This study aimed at investigating these benefits for a consecutive sample of adult patients. Improvements in speech-in-noise recognition after CI2 were followed up longitudinally for 12 months with the internationally comparable Finnish matrix sentence test. The test scores were statistically significantly better for BICI than for either CI alone in all assessments during the 12-month period. At the end of the follow-up period, the bilateral benefit for co-located speech and noise was 1.4 dB over CI1 and 1.7 dB over CI2, and when the noise was moved from the front to 90 degrees on the side, spatial release from masking amounted to an improvement of 2.5 dB in signal-to-noise ratio. To assess subjective improvements in hearing and in quality of life, two questionnaires were used. Both questionnaires revealed statistically significant improvements due to CI2 and BICI. The association between speech recognition in noise and background factors (duration of hearing loss/deafness, time between implants) or subjective improvements was markedly smaller than what has been previously reported on sequential BICI in adults. Despite the relatively heterogeneous sample, BICI improved hearing and quality of life.

## 1. Introduction

Bilateral cochlear implantation is becoming a standard of care in the treatment of bilateral severe-to-profound hearing loss, not only for pediatric patients but also for adult patients. Over more than a decade, several studies have investigated the audiological benefits, self-reported improvements, and cost-effectiveness of bilateral cochlear implants (BICI) when compared to unilateral implantation. These findings are summarized in systematic literature reviews [1,2,3] and typical bilateral benefits in adult cochlear implant (CI) recipients include improvements in speech recognition in noisy environments and more accurate sound localization compared to unilateral implantation [4,5]. Recent evidence also suggests that BICI are potentially cost-effective in adults as well as in children [3]. In a randomized clinical trial, simultaneous BICI were found to be a cost-effective treatment for patients whose life expectancy is longer than 10–25 years [6].

In addition to bilateral benefits, potential detrimental effects of CI surgery need to also be considered in patient counseling. While the overall clinical effect of unilateral CI surgery on balance and vestibular function is generally small [7], there is a higher risk of postoperative subjective vertigo when both ears have been implanted [8]. As vestibular dysfunction is prevalent in severe-to-profound hearing loss, protocols to minimize the risk for vestibular loss due to CI surgery have been proposed [9]. In contrast to surgery, electrical stimulation from the CI has, however, been suggested to improve vestibular function [10].

In the clinical routine, adult patients are often implanted sequentially with BICI while simultaneous implantation is more common in children born with severe-to-profound hearing loss. Early simultaneous BICI in pediatric cases aims to promote hearing development in both ears, to provide benefits of binaural hearing, and to prevent abnormal asymmetric strengthening and preference for the first-implanted ear associated with a long delay between the first (CI1) and the second (CI2) implant [11]. In adults, however, hearing results between simultaneous and sequential BICI have been reported to be comparable [12]. Despite positive outcomes, considerable inter-individual variability in sequential BICI exists. Several background factors affecting the bilateral outcome have been proposed, such as the time between implants [13], the duration of deafness in the CI2 ear [14], or the score obtained with the CI1 ear in adults aged over 50 years old [15]. Overall, as an increasing number of candidates with increasingly varying background factors are being considered for CI2, further clinical research on sequential BICI is warranted.

In Finland, bilateral cochlear implantation commenced first in pediatrics around 2010, and, subsequently, children have received simultaneous implants in a single surgery. Based on an individual candidacy assessment, sequential BICI in working-age adults followed some years later and the first outcomes were published soon thereafter [16]. Since then, there has been considerable development in Finnish hearing diagnostics with the nationwide adoption of the Finnish Matrix Sentence Test (FMST; [17]) and its applications [18,19,20]. The test has been developed according to international guidelines and such tests are available in a growing number of different languages, allowing for internationally comparable speech recognition thresholds (SRTs) in noise across studies [21,22]. Recently, statistically significant improvements over a follow-up period of 12 months were reported for normally hearing subjects with repeated use of the FMST [20]. It is not yet clear whether such learning effects also apply to CI users. However, possible long-term improvements due to learning should be accounted for when assessing improvements due to an intervention.

In addition to inter-individual differences in outcomes, another source of variability may stem from the physical configuration of the speech and noise signals around the test participant. While the speech signal is typically presented from the front, the noise may be co-located with the speech, it may be emanating from a different direction or hemifield than the speech, or it may be diffuse, reaching the listener with a seemingly random phase from multiple locations. These differences make direct comparisons of speech-in-noise data difficult [14]. To compare bilateral and unilateral CI hearing abilities, the spatial configuration and the test conditions can be set up such that the acoustic phenomenon of head diffraction (or head shadow) can be separated from the binaural phenomena of squelch and redundancy by comparing the obtained SRTs in noise of different configurations and conditions [23,24]. While there is a measurable benefit of head shadow with BICI, the binaural effects are generally smaller or sometimes even negligible (see, e.g., [25] for a review). The benefits of BICI are often also assessed via spatial release from masking (SRM), which is manifested as a decreasing SRT in noise with an increasing angle separation between the speech and the noise signals [24]. In normally hearing adults, SRM is largest when the speech and the noise signals emanate from different hemifields rather than from the same hemifield [26]. However, with BICI, SRM is markedly less than in normal hearing [25].

This prospective study assessed the benefits of sequential BICI for a consecutive sample of Finnish working-age adult CI recipients. The objective benefit was evaluated using the FMST in a relatively simple and clinically feasible spatial configuration of the speech and the noise signals to assess hearing within the first 12 months after receiving CI2. Improvement in SRT in noise was also assessed in the CI1 ear and compared with the observed improvement in normally hearing adults over the same period. Questionnaires were used to assess the subjective benefits of sequential BICI in this population. All adult patients eligible for CI2 were included in the study, and despite large differences in background factors, significant benefits due to BICI were identified.

## 2. Materials and Methods

### 2.1. Patients

Prior to the commencement of sequential bilateral implantation in adults in our clinics, BICI were limited to the pediatric population and special cases in adults, such as deafblindness or meningitis. We originally included the first 33 working-age adult patients eligible for BICI in this study. Four patients were pre-lingually deafened exhibiting poor development of speech recognition in their CI2 ear and they were excluded from this study. In addition, data collection was not completed for two other patients. Thus, 27 patients in total were included, and at the time of the CI2 surgery, their mean age was 44.2 (range 19.6–64.8) years and the mean duration of hearing loss (DHL) and duration of deafness (DD) in the CI2 ear were 35.1 (range 18.2–56.2) and 21.0 (range 1.8–47.9) years, respectively. DHL and DD were computed from the patient records for each ear and defined as the time between the CI surgery and when the pure-tone average (PTA; average hearing threshold of 500, 1000, 2000, and 4000 Hz) in that ear had been worse than 25 and 80 dB hearing level (HL), respectively. The mean time between the implantation of CI1 and CI2 was 5.4 (range 1.0–16) years. The mean (± standard deviation; SD) preoperative PTA in the CI2 ear was 99.1 ± 12.4 dB HL. Of the 27 patients, eight used a hearing aid (HA) up until the CI2 surgery.

A soft surgical technique was utilized in the CI2 surgery [27]. The surgery was performed via a transmastoid/posterior tympanotomy approach with the electrode inserted through the round window or antero-inferiorly extended round window into the scala tympani. A total of nineteen patients were implanted with a Med-El and eight patients with a Cochlear device. A lateral-wall electrode array was used for 25 patients and a perimodiolar array for two patients. The implant brand and type together with patient etiology and background factors are listed in Table 1. Directional microphones that could affect the SRTs in noise were disabled in the CI sound processors during the sound-field testing.

The ethical committee of Helsinki University Hospital approved the study protocol (approval code 113/13/03/02/2015). All patients gave their voluntary informed consent to participate in the study.

### 2.2. Test Apparatus

Speech recognition in noise was measured in a soundproof listening room over two loudspeakers (Genelec 8050A, Genelec, Iisalmi, Finland). The loudspeakers were located at ear level and at a distance of 1 m from the listening position at a 90° angle in between them. Post-operative sound-field hearing thresholds were measured separately for CI1 and CI2 according to the provisions of ISO 8253-2 and ISO 389-7 standards [28,29] with a loudspeaker in front of the patient.

The SRTs in noise for the FMST were measured with the Oldenburg Measurement Applications test software (Hörtech, Oldenburg, Germany), a laptop computer, and an Auritec ear 3.0 sound card in three different configurations: speech and noise co-located in the front (S0N0), speech in the front and noise either to the left or right of the patient (S0N − 90 and S0N + 90, respectively). The speech material consisted of equally intelligible test lists, and each list comprised 20 sentences. The noise was stationary with the same long-term spectrum as the speech signal.

### 2.3. Procedure

After switching CI2 on, sound-field hearing thresholds were measured when needed, to ensure that the audiograms for CI1 and CI2 were equal or better than 30 dB HL in the audiometric frequencies from 250 Hz to 6 kHz. Adjustments to CI fitting were done accordingly to achieve this.

The SRTs in noise were measured preoperatively with CI1 and a potential HA at 1, 3, 6, and 12 months after switching CI2 on. Five test configurations were used: three in the S0N0 condition (CI1 only, CI2 only, BICI), and bilaterally in the S0N −90 and S0N + 90 conditions, where the noise signal was either on the side of the first or the second implanted ear (S0NCI1 and S0NCI2, respectively).

The overall level of the noise was fixed at 65 dB SPL. Each session started with two practice lists; the first list of sentences was administered with a fixed + 10 dB signal-to-noise ratio (SNR) and the second starting from 0 dB SNR and varying the SNR in an adaptive procedure to converge to 50% correct speech recognition, i.e., to SRT in noise. After the two practice lists, the five configurations were administered in a counterbalanced order across sessions and patients to minimize the effect of within-session fatigue. The test list for each SRT-in-noise measurement was selected at random, and the word matrix was presented in writing to the patients at the beginning of each test session to minimize the effect of content learning of the matrix sentence material.

The impact of CI2 on quality of life (QoL) was measured using the Glasgow Benefit Inventory (GBI) and the Glasgow Health Status Inventory (GHSI). GBI is a validated, generic patient-recorded outcome measure widely used in otolaryngology to report changes in QoL after an intervention [30]. Both questionnaires were translated from English to Finnish, and they were self-completed by the patients answering 18 questions in each questionnaire using a five-point Likert scale. GBI addressed the change in health status 12 months after switching CI2 on. The individual responses were then scaled and averaged to give a score within a range −100 (poorest outcome) through 0 (no change) to + 100 (best outcome). GBI resulted in a total score and three subscale (general, 12 questions; social, 3 questions; physical, 3 questions) scores. GHSI focuses on hearing difficulties and their impact on daily life at the time the questionnaire was completed. GHSI allowed for comparisons between time points prior to receiving CI2 and 12 months after switching CI2 on. Similar to GBI, GHSI resulted in a total score and three subscale scores. Each subscale score was calculated as follows: (subscale score − 1) × 25. A higher score indicated better health status and a lower score indicated poorer health status for each subscale.

### 2.4. Statistical Analyses

A nonparametric Friedman’s test of differences among repeated measures was performed on the SRT-in-noise data separately for the CI1, CI2, and BICI in the S0N0 condition, and for the bilateral S0N0, S0NCI1, and S0NCI2 conditions using the IBM SPSS Statistics Version 22 software (IBM, Armonk, NY, USA). The GBI and GHSI scores were analyzed using Student’s t-test in Graphpad Prism 8 (GraphPad Software, San Diego, CA, USA). Pearson’s correlation coefficients were computed for the SRT-in-noise data, the background factors of Table 1, and the questionnaire data with the Matlab Version R2018a software (MathWorks, Natick, MA, USA).

## 3. Results

### 3.1. Audiometric Outcomes

Post-operative PTA (mean ± SD) in the CI2 ear was 35.8 ± 10.5 dB HL after the switch on and 31.5 ± 5.4 dB HL at 1–2 months from switching CI2 on. For two patients, the audiogram after the switch on was not available. The patients were instructed to wear both the CI1 and CI2 sound processors in their everyday life and occasionally to practice their listening abilities with CI2 only.

Based on the Shapiro–Wilk test, the preoperative and some of the 1-, 3-, and 6-month SRT-in-noise data were not normally distributed (*p* < 0.05) and, therefore, the nonparametric Friedman’s test was performed on the data. For CI1, CI2, and BICI in the co-located S0N0 condition, Friedman’s test produced an χ2 value of 166.83, and for the bilateral S0N0, S0NC1, and S0NCI2 conditions an χ2 value of 106.73, which were both significant (*p* < 0.001).

Figure 1 shows the median SRT in noise in the S0N0 condition 1, 3, 6, and 12 months after switching CI2 on, separately for CI1, CI2, and BICI, as well as the preoperative SRT in noise with CI1 and a potential HA. A Wilcoxon signed-rank test was performed post hoc and thirty-four pairwise comparisons were made: (1) between each assessment point in time (1, 3, 6, and 12 months) for each listening mode (CI1, CI2, and BICI) resulting in 6 × 3 comparisons, (2) within each assessment point between each listening mode resulting in 4 × 3 comparisons, and (3) between the preoperative SRT in noise and each assessment point for CI1 resulting in additional 4 comparisons. A Bonferroni adjustment for multiple comparisons was made by dividing the significance level α by the total number of tests (*N* = 34). SRTs in noise were statistically significantly better for BICI than for CI1 (*p* < 0.00016) and for CI2 (*p* < 0.00002) at each post-operative assessment point. One month after switching CI2 on, the SRT in noise was better for CI1 than for CI2 (*p* = 0.00076). These differences are marked with asterisks on the corresponding significance level in Figure 1. Twelve months after switching CI2 on, the median SRTs in noise were −5.1, −4.8, and −6.5 dB SNR for CI1, CI2, and BICI, respectively, while it was −3.9 dB SNR preoperative to CI2 for the patients of the present study.

As is evident in Figure 1, there is considerable variation in the results across patients and SRT in noise appears to be improving over time. The improvement in the median SRT in noise over the follow-up period was significant for CI2 between 1 and 3 months (Δ = 1.5 dB SNR; *p* = 0.00013), 1 and 6 months (Δ = 2.1 dB SNR; *p* = 0.00001), 1 and 12 months (Δ = 2.7 dB SNR; *p* < 0.00001), and 3 and 12 months (Δ = 1.2 dB SNR; *p* = 0.00015). For BICI, the improvement was significant between 1 and 6 months (Δ = 1.2 dB SNR; *p* = 0.00017) and 1 and 12 months (Δ = 1.1 dB SNR; *p* = 0.00022). Finally, the SRT in noise between the preoperative CI with a potential HA and the post-operative CI1 improved significantly 6 and 12 months after switching CI2 on (Δ =1.1 dB SNR; *p* = 0.00067 and Δ = 1.2 dB SNR; *p* = 0.00019, respectively).

Figure 2 shows the median SRT in noise for speech in the front and noise either co-located with the speech signal (S0N0, BICI in Figure 1) or the noise at 90° on the side of the first (S0NCI1) or the second (S0NCI2) implanted ear. In addition, an estimated CI1-only performance is shown in the S0NCI1 condition based on the mean preoperative SRT of the present study and the difference between S0N0 and S0NCI1 in listening via a unilateral CI [19]. Similar to the data presented in Figure 1, the Wilcoxon signed-rank post-hoc test was utilized and pairwise comparisons were made: (1) between each assessment point in time (1, 3, 6, and 12 months) for the listening modes with the noise on the side (S0NCI1 and S0NCI2) resulting in 6 × 2 comparisons, and (2) within each assessment point between each listening mode resulting in 4 × 3 comparisons, and a Bonferroni adjustment was made to the total number of tests (N = 24). The difference between S0N0 and S0NCI1 or S0NCI2 was significant at each post-operative assessment point (*p* < 0.00050; marked with asterisks in Figure 2), except between S0N0 and S0NCI1 1 month after switching CI2 on. At the end of the follow-up period, the median SRT in noise both for S0NCI1 and S0NCI2 conditions was −9.0 dB SNR, which is 2.5 dB lower than in the co-located S0N0 condition.

In Figure 3, the individual SRTs in noise after 12 months from switching CI2 on are plotted against the background factors listed in Table 1. The duration of hearing loss (DHL; N = 24) is plotted as a factor in panels A–E, the duration of deafness (DD; N = 25) in the implanted ear in panels F–J, and the time between CI1 and CI2 surgeries (time between implants, TBI; N = 27) in panels K–O. Since listening is only based on the CI1 ear in panels A and F and mostly on it in panels E and J with the noise on the side of CI2, DHL and DD on the abscissae of these panels are expressed for the CI1 ear. The abscissae in panels B, D, G, and I are expressed for the CI2 ear. In panels C and H, they are expressed as a mean of the CI1 and CI2 ears. The association between SRT in noise and any of the background factors is moderate at best and statistically significant only in panel A, i.e., the longer the DHL in the CI1 ear, the higher the SRT in noise indicating a poorer hearing outcome when listening with CI1 only (Pearson’s r = 0.43, *p* < 0.05). There is a moderate association (|r| > 0.3; panel M of Figure 3) between bilateral SRT in noise in the S0N0 condition and time between implants; however, this did not reach statistical significance (*p* = 0.11). For the rest of the data in Figure 3, the absolute value of Pearson’s correlation coefficients is below 0.3, suggesting a small or lack of association between the SRTs in noise and the background factors. These correlation coefficients were also computed for the 1-, 3-, and 6 month data and they were statistically significant for CI1 and BICI in the S0NCI2 condition for the DHL background factor 1 and 3 months after switching CI2 on (corresponding to panels A and E of Figure 3; Pearson’s r = 0.41–0.48, *p* < 0.05). This further bolsters the association between the duration of hearing loss and poorer hearing outcomes in the CI1 ear. However, no such association was found for CI2 or BICI in our consecutive patient sample.

### 3.2. Questionnaire Data

To measure the impact of CI2 on the patients’ QoL, the GBI was self-completed by the patients (N = 27) 12 months after switching CI2 on (Figure 4A). One patient did not return the questionnaire. The CI2 had a widespread effect on QoL as the total GBI score was + 55 ± 4% (scale from −100 to + 100, mean ± SEM), general subscale + 71 ± 4%, social subscale + 31 ± 7%, and physical subscale + 17 ± 6%, all of which statistically differed from the hypothetical value of 0 corresponding to no change after intervention.

In comparison to the GBI, GHSI is more focused on hearing difficulties and their impact on daily life. It allows comparison between two time points. In this study, the GHSI was self-completed by the patients (N = 23) prior to receiving CI2 and 12 months after switching CI2 on. Five patients failed to return the questionnaire either once or twice. When the GHSI scores were compared between prior to CI2 and 12 months after the CI2 switch on, the total score was higher 12 months after the CI2 switch on than prior to CI2 surgery (66 ± 2 vs. 61 ± 2, respectively; *p* < 0.05; Figure 4B). The general subscale score was also significantly higher after the CI2 switch on than prior to CI2 surgery (64 ± 3 vs. 56 ± 2, respectively; *p* < 0.05). There were no differences in the social support or physical health subscales between the time points.

Finally, the SRT-in-noise improvement from the preoperative CI and a potential HA to BICI 12 months after switching CI2 on is plotted against the total GBI CI2 and the three subscale scores in Figure 5 with the corresponding linear regressions and the respective Pearson’s correlation coefficients and their p-values. In all but the General subscale (panel B in Figure 5), there is a statistically significant positive association between the SRT-in-noise improvement and the GBI CI2 score.

## 4. Discussion

Improvements in SRT in noise, in hearing, and in QoL were investigated in this study for a consecutive sample of sequentially implanted deafened adult patients over a follow-up period of 12 months from switching CI2 on. In the co-located S0N0 condition (see Figure 1), there was a statistically significant advantage of bilateral over unilateral SRT in noise in each post-operative assessment, and at the end of the follow up the SRT in noise for BICI was 1.4 and 1.7 dB lower than for CI1 and CI2, respectively. This advantage is referred to as binaural redundancy or diotic summation in the literature [24,25], and similar advantages have been reported previously for BICI in adults [23,25]. As the test conditions of the present study were limited by what was feasible in our clinical routine, the amount of head shadow or binaural squelch cannot be directly estimated from the present data.

When looking at the longitudinal improvement in SRT in noise, the median CI2 performance reached that of CI1 3 months after switching CI2 on. Again, this is in line with earlier research on CI2 improvement over time in sequential BICI in adults [14]. In addition to Figure 1, where the difference between CI1 and CI2 becomes insignificant at 3 months, this can also be seen in the bilateral data of Figure 2 where the difference from the S0N0 to the S0NCI1 and S0NCI2 conditions becomes significant at the same assessment point. When the noise signal is on the side of CI1, listening is based more on the speech signal reaching CI2 while the noise on that side is being shadowed by the patient’s head. In later assessments at 6 and 12 months, there were no statistically significant differences in the unilateral CI1 and CI2 nor in the bilateral S0NCI1 and S0NCI2 conditions, despite the fact that our consecutive sample contained relative heterogeneous SRT-in-noise data, at least when compared to the Finnish validation data for FMST in unilateral CI recipients [18].

In clinical practice, it is often assumed that speech audiometric results are stable unless there are changes in hearing. However, learning effects are acknowledged for most sentence-based speech in noise tests. While within-session learning in SRT-in-noise measurements is also well documented for the multilingual matrix test [22] and for its Finnish version (FMST; [17]), less has been reported about the long-term, inter-session learning of these tests [20]. To minimize the effect of within-session improvement on our SRT-in-noise data, we counterbalanced the order of the different test conditions for the FMST in the present study. However, we still noted a statistically significant improvement of the CI1 ear over the follow-up period of 12 months (see Figure 1). We assumed that prior to the CI2 surgery, the hearing outcome of CI1 had, at least on average, stabilized, because the mean time between implantations was 5.4 years (see Table 1). Some of this improvement could be due to changes in CI1 fitting at the post-operative fitting sessions of CI2, or potentially even due to a detrimental effect of the contralateral HA at the preoperative assessment. However, only eight out of the 27 subjects used a HA up until the CI2 surgery so the latter effect is unlikely.

To explore the inter-session improvement in more detail, we compared the median SRT in noise for CI1 of the present study with the mean of normally hearing listeners from [20], both assessed at the same time points after their respective first exposure to the FMST. As the presentation order of CI1 was equally distributed along the measurements within a session in this study, we averaged the four measurements (M1–M4) in [20] to represent an average SRT in noise for each session. Even though there are differences in the statistical distributions of these data, the improvement in the median values of CI1 resembles the mean improvement observed in [20]. At 12 months after the first exposure, the improvement was 1.2 dB and 1.3 dB for CI and normal hearing, respectively. However, as we did not control for the effect of fitting changes in CI1 in this study, more studies are needed on the long-term improvements to make further inferences. This is especially important for longitudinal studies with repeated testing and exposure to speech tests in noise, as well as for telemedicine and self-testing applications.

In normally hearing listeners, SRM has been reported to be up to and above 10 dB when SRT is compared between co-located and spatially separated speech and noise signals [23,31,32]. In BICI, this effect is considerably smaller, in the order of 3 dB [25], when investigated with the speech signal in the front and the noise signal on the side. In this study, SRM was 2.5 dB at the end of the follow-up period, which is in agreement with the literature. Overall, the bilateral advantage resulting in binaural redundancy and SRM and the improvement of CI2 over time agree well with earlier research, irrespective of the relatively large individual differences in preoperative SRTs in noise in our consecutive patient sample.

The background factors, however, correlated fairly little with the SRT in noise data in the present study. Patient age or etiology did not seem to influence the audiometric outcome in our study sample: SRTs in noise were close to the median results for the oldest patients, as well as for patients with genetic hearing loss. For 14 out of 27 patients, however, the etiology of hearing loss was unknown. In an earlier longitudinal study of sequential bilateral CIs in adults [14], the DD of the CI2 ear was strongly associated with the hearing outcome for CI1, CI2, and BICI in a number of different speech tests both in quiet and in noise with an absolute value of the correlation coefficient being equal or greater than 0.6. A similarly strong association has been published between TBI and the difference between the speech scores of CI1 and CI2 in quiet [13]. In our study, the background factors were associated with the hearing outcome only for CI1 and to a much lesser degree. There were no marked differences in etiology or in the background factors between the studies, with both the range and the mean of TBI and DD in the CI2 ear being very similar (TBI range 1.0–16, mean 5.4; DD CI2 range 1.8–47.9, mean 21.0 years in the present study vs. TBI range 1.0–16, mean 5.2; DD CI2 range < 1—55, mean 21.4 years in [14]). Some of the background factors were not available in our patient records or were potentially inaccurate, and this may have lowered the associations presented here. As there could be differences in, e.g., patient enrolment, further research is needed to identify factors contributing to hearing outcomes, especially with regard to CI2 and BICI in consecutive patient series.

Generic QoL instruments typically underestimate the impact of BICI. Meta-analysis with hearing or CI-specific QoL instruments shows large improvements in patients with BICI and medium improvements when comparing bilateral to unilateral CI [33]. Our results are in agreement with these findings. In our study, the impact of CI2 on QoL was evaluated with the GBI and GHSI questionnaires. The GBI questionnaire was originally developed to evaluate the impact of interventions on QoL [30] and it has been used widely for this purpose. GBI has been used to evaluate the benefits of sequential BICI [16]. BICI increased patients’ working performance and decreased work-related stress. Patients were more active in their working environment and communication with co-workers was easier. Again, sequential bilateral cochlear implantation improved QoL [16]. In our study, the GBI total score and all the subscale scores were positive 12 months after the switch on, suggesting that receiving CI2 was beneficial. The GHSI questionnaires were done at two different time points. Statistically significant changes were seen in the total score and in the general subscale score. Social and physical subscale scores did not reveal any significant difference. With more hearing-specific questionnaires, larger associations with speech-in-noise test results have been reported for BICI [34]. Thus, further research with similar patient populations may reveal whether positive changes in QoL are associated with improvements in hearing.

## 5. Conclusions

Statistically significant improvements in hearing and in QoL were obtained over a follow-up period of 12 months for the consecutive patient sample of this study. The benefit of BICI was consistent with the literature. The background factors were only moderately associated with speech recognition in noise and this was the case only for the CI1 ear. For CI2 or BICI, the association was insignificant, which is inconsistent with recent research on a similar study sample. A significant improvement over the follow-up period was also noted for CI1, to the same extent as has been reported in repeated testing for normally hearing listeners. Finally, the association between a generic QoL questionnaire and speech recognition in noise was moderate.

## Figures and Tables

**Figure 1 jcm-10-02394-f001:**
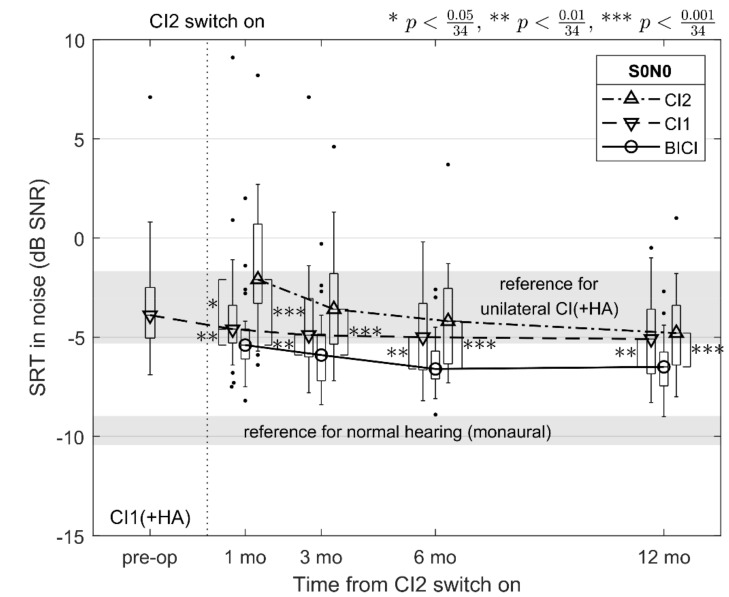
Median SRTs in noise preoperatively for the first implanted ear (CI1) and a potential hearing aid (HA) in the contralateral ear, and at 1, 3, 6, and 12 months (mo) post-operatively after switching on the second implanted ear (CI2), measured with each implant separately (CI1 and CI2) and bilaterally with both implants (BICI) with the speech and noise signals co-located in the front (S0N0). The boxplots denote the interquartile range, their whiskers the 2nd and the 98th percentile, and the dots represent values outside this range. The shaded areas depict the expected ranges (mean ± 1 SD) for a normally hearing person listening monaurally over headphones ([17];−9.7 ± 0.7 dB SNR) and for a unilateral Finnish CI recipient ([18]; −3.5 ± 1.7 dB SNR).

**Figure 2 jcm-10-02394-f002:**
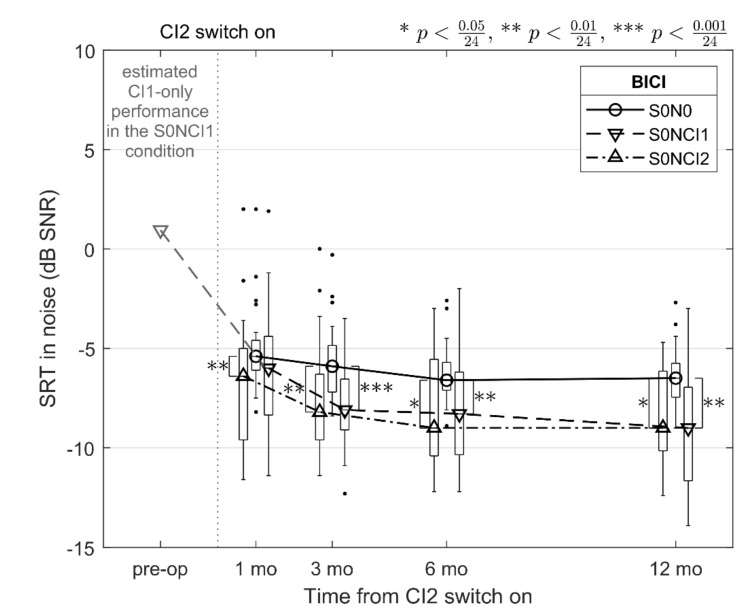
Median SRTs in noise 1, 3, 6, and 12 months after switching CI2 on, measured bilaterally (BICI) with the speech and noise co-located in the front (S0N0), speech in the front and the noise at a 90° angle on the side of CI1 or CI2 (S0NC1 and S0NCI2, respectively). The boxplots denote the interquartile range, their whiskers the 2nd and the 98th percentile, and the dots represent values outside this range. The light-gray downward-pointing triangle in the left of the figure is an estimated SRT in noise in unilateral CI listening of a frontal speech signal with the noise emanating at a 90° angle on the side of the CI sound processor, based on the preoperative SRT of the present study and the difference between S0N0 and S0NCI in [19].

**Figure 3 jcm-10-02394-f003:**
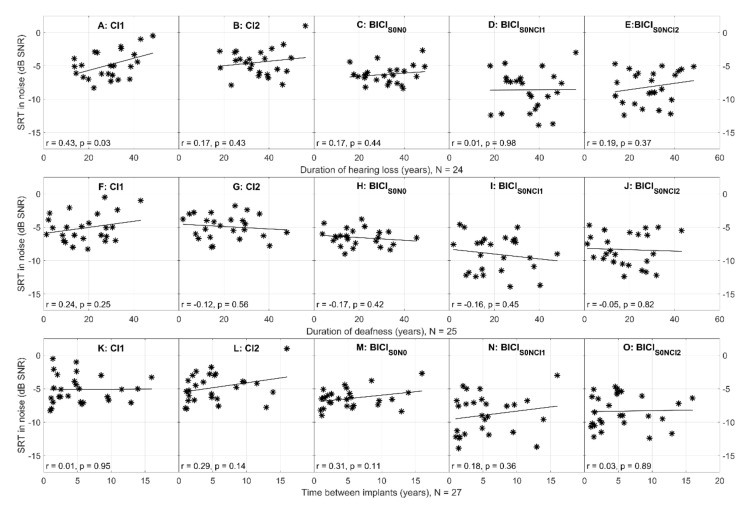
Individual SRTs in noise 12 months after switching CI2 on plotted against the duration of hearing loss (DHL), duration of deafness (DD), and time between implants (TBI) in years. In panels (**A**), (**E**), (**F**), (**J**), DHL, and DD are expressed for CI1 ear, and in panels (**B**), (**D**), (**G**)**,** and (**I**), for CI2 ear. In panels (**C**) and (**H**), they are expressed as a mean of CI1 and CI2. The linear regression is plotted as a solid line with the corresponding Pearson’s correlation coefficient and the respective *p*-value in the lower-left corner of each panel.

**Figure 4 jcm-10-02394-f004:**
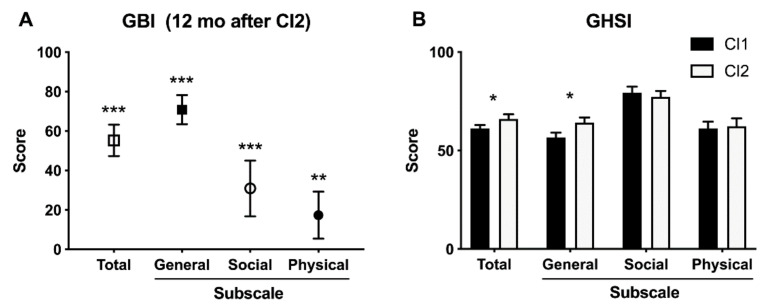
The impact of CI2 on the quality of life. (**A**) GBI scores 12 months after switching CI2 on. Data are mean ± SEM (N = 27). **, *p* = 0.0062; ***, *p* < 0.0001 for the difference from hypothetical value of 0 (Student’s t-test). (**B**) GHSI scores prior to CI2 surgery and 12 months after switching CI2 on. Data are mean ± SEM (N = 23). *, *p* < 0.05 for the difference between time points (Student’s t-test).

**Figure 5 jcm-10-02394-f005:**
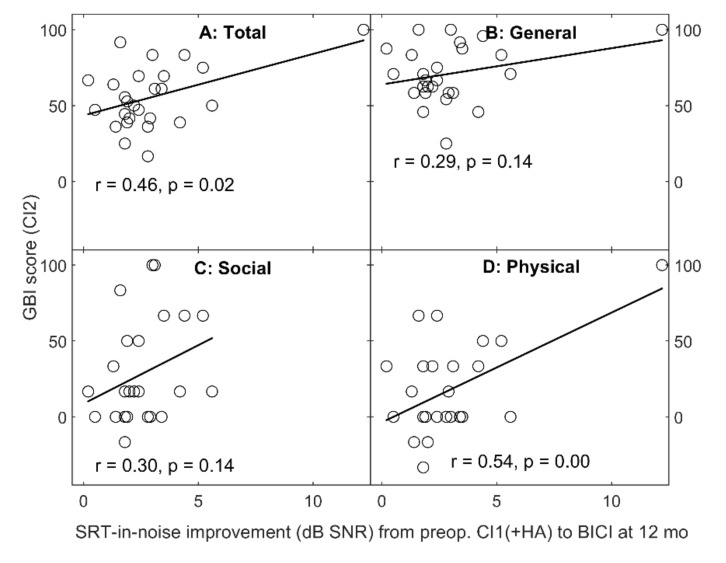
The improvement of SRT in noise from the preoperative CI1 (with a potential HA) to BICI 12 months after switching CI2 on plotted against the total GBI CI2 score (panel **A**) and the three subscales (General, Social and Physical, panels **B**–**D**). The linear regression is plotted with a solid line with the respective Pearson’s correlation coefficient (r) and the p-value.

**Table 1 jcm-10-02394-t001:** Etiology, CI brand and type, and background factors of the patients.

						Time betweenImplants	Duration of Hearing Loss	Duration of Deafness
	Age							
Patient	(Years)	Brand	Etiology	CI1	CI2	(Years)	CI1/CI2 (Years)	CI1/CI2 (Years)
1	58.6	Med-El	unknown	C40 + Standard	Concerto FLEX28	13.9	29.8/43.7	10.2/24.2
2	39.3	Cochlear	unknown	CI512	CI512	4.9	34.4/39.3	32.7/37.6
3	41.1	Cochlear	unknown	CI24RE	CI512	5.3	31.1/36.4	30.5/35.8
4	52.5	Med-El	unknown	C40 + Standard	Concerto FLEX28	12.9	33.1/46.0	27.3/40.3
5	35.2	Med-El	Usher	Concerto FLEX28	Concerto FLEX28	1.1	30.5/31.7	27.9/29.4
6	35.3	Med-El	Usher	Concerto FLEX28	Synchrony FLEX24	1.2	unknown	12.9/14.6
7	55.8	Med-El	meningitis	Pulsar ci100 Standard	Synchrony FLEX28	8.5	23.9/32.4	23.9/18.4
8	47.9	Med-El	rubella	Sonata ti100 Standard	Synchrony FLEX28	4.8	43.1/47.9	43.1/47.9
9	34.5	Cochlear	genetic	CI422	CI422	2.0	22.7/24.7	2.7/4.7
10	48.9	Med-El	unknown	Sonata ti100 Standard	Concerto FLEX28	4.8	41.6/46.4	20.1/24.9
11	35.2	Med-El	unknown	Concerto Standard	Synchrony FLEX24	3.6	14.3/31.2	1.3/28.3
12	39.0	Med-El	Usher	Concerto FLEX28	Synchrony FLEX24	2.3	unknown	8.4/8.7
13	56.2	Med-El	unknown	C40 + Standard	Synchrony FLEX24	16.0	40.2/56.2	unknown
14	55.4	Med-El	unknown	Sonata ti100 FLEX28	Concerto FLEX28	4.5	13.6/18.2	2.1/6.7
15	27.3	Med-El	genetic	Sonata ti100 Standard	Synchrony FLEX24	5.5	19.9/25.4	8.9/14.4
16	56.3	Med-El	unknown	C40 + Standard	Synchrony FLEX28	11.5	13.7/25.2	4.0/15.6
17	19.6	Cochlear	unknown	CI24RE(CA)	CI522	5.9	unknown	unknown
18	23.3	Med-El	genetic	Concerto FLEX24	Synchrony FLEX28	1.0	22.3/23.3	19.6/15.1
19	39.1	Med-El	sudden	Concerto FLEX28	Synchrony FLEX28	2.5	25.7/29.7	25.5/29.5
20	49.2	Med-El	unknown	Concerto FLEX28	Synchrony FLEX28	2.6	38.7/41.2	27.7/30.3
21	51.9	Cochlear	Cogan	CI24RE(CA)	CI422	9.6	17.6/27.2	17.6/12.4
22	35.5	Med-El	meningitis	Concerto FLEX28	Synchrony FLEX28	1.5	16.9/18.4	16.9/11.8
23	49.8	Med-El	CMV	Concerto FLEX28	Synchrony FLEX28	1.4	48.5/49.8	27.1/1.84
24	44.7	Cochlear	unknown	CI422	CI522	1.4	38.3/39.6	31.9/27.0
25	64.8	Med-El	otosclerosis	Concerto FLEX24	Concerto FLEX24	1.5	34.3/35.8	11.5/7.5
26	45.6	Cochlear	unknown	CI24RE(CA)	CI522	9.5	28.7/38.2	13.7/23.1
27	52.6	Cochlear	unknown	CI512	CI522	5.7	29.8/35.5	9.6/13.9
Mean	44.2					5.4	28.9/35.1	18.3/21.0

## Data Availability

The data presented in this study are available on request from the corresponding author. The data are not publicly available due to information that could compromise the privacy of research participants.

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
