# Peer review of "Improvements in Hearing and in Quality of Life after Sequential Bilateral Cochlear Implantation in a Consecutive Sample of Adult Patients with Severe-to-Profound Hearing Loss"

_jcm, 2021, doi:10.3390/jcm10112394_

Round 1

Reviewer 1 Report

The authors propose a study on the evaluation of the benefit of sequential bilateral cochlear implantation in adults, in terms of quality of life and hearing performance in noise. Overall, the article is clear and well written.

My comments will be limited to two main points:

1) The topic is very interesting but has already been widely explored in the literature, and many data already indicate the benefit of bilateral implantation. This article provides additional data on a series of patients, but no innovative information. I think that the data lack originality, even if they confirm many other studies on this topic.

2) My second comment is about the methodology. The description of the overall design of the study lacks precision, especially concerning the retrospective or prospective nature of the data, which is not clearly mentioned. It seems that most of the data were collected retrospectively from the medical records. For example, there was significant missing data on audiometric tests (data available for 17 out of 27 patients), which weakens the power of the data.

The discussion shows some interesting analyses, in particular on a potential long-term learning effect with audiometric tests in noise.

Reviewer 2 Report

The article refers to the very important problem of quality of life in adult patients after sequentially performed bilateral cochlear implantation. 

As it have been reported in many papers the advantage of bilateral cochlear implantation may occur due to improvement of speech recognition in a difficult environmental situations, , and improvement of sound localisation.

The study rightly showed statistically significant improvement in hearing and quality of life in  adult patients using bilateral cochlear implants as compared with single implant use. 

The methodology of audiometric measurements is reasonable and proper as well as Glasgow Benefit Inventory is the internationaly well recognised tool for measurements of QQL after surgical and medical  interventions. 

However I have few comments to the article mostly about things which have not been mentioned in the study and which decrease the value of the paper. 

There is a general agreement in adult population that congenital deafness usually is not very good situation for cochlear implantation because of hearing deprivation. Especially when the patient did not use hearing aid.  However there is mounting evidence that implanation can also be beneficial in this situation.

Another case  is when the patient has progressive hearing loss and decreasing discrimination – here CI usually gives very good results. So I would seek in this paper referal to this problem.

Nothing is mentioned that depending on the  audiometric results – some of the patients might  benefit also from bimodal stiumulation before second CI and this patients if using hearing aid on contralateral site have also better  QQL and may benefit from bilateral hearing. The difference between  bimodal stimulation and bilateral cochlear implants may be much less that comparison of single implant with two implants. How many patients were using HA contralaterally before second CI. The comparison of this two groups will be also very important.

Very  important factor about quality of life of bilateral CI recipients is the fact that it is well established that cochlear implantation may impair vestibular function via direct injury or inflamation. Vienner Vacher  and  and many others reported vestibular hypofunction in 50  % of the cochlear implant recipients and total vestibular loss in 10 % of patients. Unilateral vestibular areflexia with normal contralateral function should not be a clinical problem for the patients due to compensatory processes. However bilateral complete bilateral loss of vestibular function may pose a significant problem for motor function especially in adult -  elderly patients where compensatory processess may be delayed. This situation is fortunately not common but if present  may pose significant problem – huge balance problems that may be irreperable . I have seen it few times.

Vestibular testing in small children may be very difficult , the centers using bilateral simultaneous implantation  protocol has to accept 1-2 % of total bilateral vestibular loss due to implantation with subsequent poor motor development in these children. But in adults  vestibular testing  should not be a problem and is rather mandatory especially in sequential cochlear implantation.

Nothing is written whether the authors checked vestibular system (caloric test) to asses possible  vestibular function loss in an already implanted ear. If so the other ear should better not be implanted because of the risk of  total bilateral vestibular  areflexia which may have huge negative effect on balance and motor fuction.

This should be done due to ethical reasons  in order to prevent significant deterioration in quality of life in these patients.

This  article about bilateral implantation should mention about the possible vestibular problem which is well known for CI centres and may have huge  deleterious effect on quality of life of  adult patient  with sequentially added  second implant. 

Author Response

REVIEWER 2 COMMENTS

The article refers to the very important problem of quality of life in adult patients after sequentially performed bilateral cochlear implantation. 

As it have been reported in many papers the advantage of bilateral cochlear implantation may occur due to improvement of speech recognition in a difficult environmental situations, , and improvement of sound localisation.

The study rightly showed statistically significant improvement in hearing and quality of life in  adult patients using bilateral cochlear implants as compared with single implant use. 

The methodology of audiometric measurements is reasonable and proper as well as Glasgow Benefit Inventory is the internationaly well recognised tool for measurements of QQL after surgical and medical  interventions.

However I have few comments to the article mostly about things which have not been mentioned in the study and which decrease the value of the paper. 

There is a general agreement in adult population that congenital deafness usually is not very good situation for cochlear implantation because of hearing deprivation. Especially when the patient did not use hearing aid. However there is mounting evidence that implanation can also be beneficial in this situation.

We thank the reviewer for highlighting this. For three out of 27 patient, the etiology of hearing loss was genetic. Their results, however, are close to the median values of our study sample and this did not seem to influence the outcome. As etiology was unknown for approximately half of the patients, there may be factors which remain unaccounted for in our clinical study sample. To clarify this issue, we have added discussion about patient etiology in section 4 (lines 398-401).

Another case  is when the patient has progressive hearing loss and decreasing discrimination – here CI usually gives very good results. So I would seek in this paper referal to this problem.

Nothing is mentioned that depending on the  audiometric results – some of the patients might  benefit also from bimodal stiumulation before second CI and this patients if using hearing aid on contralateral site have also better  QQL and may benefit from bilateral hearing. The difference between  bimodal stimulation and bilateral cochlear implants may be much less that comparison of single implant with two implants. How many patients were using HA contralaterally before second CI. The comparison of this two groups will be also very important.

In our patient sample, the mean preoperative PTA in the ear to be implanted (CI2 ear) was poor (close to 100 dB HL). Based on our clinical experience, such a sample is not ideal for bimodal (CI+HA) hearing rehabilitation. For the adaptive measurement of SRT in noise to be successful, we typically require 70 % correct or higher with a fixed SNR of +10 dB SNR (the first practice list of sentences, section 2.3, lines 164-165). For the vast majority of our patients, this was not possible given their PTA and/or lack of HA use. Only eight of the 27 patients wore a HA contralaterally before CI2 (section 2.1, line 123). Therefore, we believe bilateral CI to be more optimal than bimodal hearing rehabilitation for our study subjects. 

Very  important factor about quality of life of bilateral CI recipients is the fact that it is well established that cochlear implantation may impair vestibular function via direct injury or inflamation. Vienner Vacher  and  and many others reported vestibular hypofunction in 50  % of the cochlear implant recipients and total vestibular loss in 10 % of patients. Unilateral vestibular areflexia with normal contralateral function should not be a clinical problem for the patients due to compensatory processes. However bilateral complete bilateral loss of vestibular function may pose a significant problem for motor function especially in adult -  elderly patients where compensatory processess may be delayed. This situation is fortunately not common but if present  may pose significant problem – huge balance problems that may be irreperable . I have seen it few times.

Vestibular testing in small children may be very difficult , the centers using bilateral simultaneous implantation  protocol has to accept 1-2 % of total bilateral vestibular loss due to implantation with subsequent poor motor development in these children. But in adults  vestibular testing  should not be a problem and is rather mandatory especially in sequential cochlear implantation.

Nothing is written whether the authors checked vestibular system (caloric test) to asses possible  vestibular function loss in an already implanted ear. If so the other ear should better not be implanted because of the risk of  total bilateral vestibular  areflexia which may have huge negative effect on balance and motor fuction.

We thank the reviewer for pointing this out. In our clinical routine, video head impulse test (vHIT) is performed prior to CI surgery as a baseline in case postoperative problems with balance emerge. In this patient series, we did not come across clinically significant balance problems following bilateral cochlear implantation. Because we typically did not need to measure vHIT postoperatively, we are cautious not to draw any conclusions on the issue of balance based on the clinical data collected for this patient sample. Furthermore, vHIT is most likely somewhat insensitive in measuring vestibular dysfunction after CI surgery and as helpfully pointed out by the reviewer, a caloric test (or VEMP) may be required.

We have highlighted the fact that a soft surgical approach was taken in all CI2 surgeries of the present study and most patients were implanted with a lateral-wall electrode array (section 2.1, lines 125-130). However, it is speculative whether these factors truly play a role in avoiding the risk of vestibular dysfunction, and therefore, we feel we should avoid making any further inferences in the text based on the present data.

This should be done due to ethical reasons  in order to prevent significant deterioration in quality of life in these patients.

In our study sample, quality of life (as measured by GBI) did not deteriorate after CI2, but rather improved in all subscales, including the physical subscale. However, a balance-specific questionnaire (such as DHI) was not utilized in this study, and this study focuses on the audiometric and QoL outcomes.

This  article about bilateral implantation should mention about the possible vestibular problem which is well known for CI centres and may have huge  deleterious effect on quality of life of  adult patient  with sequentially added  second implant. 

Based on the reviewer’s comprehensive feedback and helpful suggestion, we have added a paragraph with literature references in the introduction (lines 45-52) where potential detrimental effects of (bilateral) CI on balance are summarized. Overall, we hope that this helps the reader get a more realistic understanding of the advantages, and potential disadvantages, of bilateral CI.

Reviewer 3 Report

Dear authors,

Overall, I have found your study very interesting. It is well written and it is scientifically sound. 

In fact, I only have one small concern about your study. First, I do like the fact that you mention that you did not control for some variables in the discussion section. However, I am wondering :

a) What is the effect of the large range in participants' age on the results? Indeed, your variables have a very large range (e.g., age, duration of deafness, etc.), but you did not control for all of them. I am mentioning age specifically because age, cognition, and cochlear implantation have been found to be somewhat related. So, here I am wondering why you did not control for age and why you did not look at cognition (in the questionnaires post implantation, for example). 

b) Is it possible to expand a bit on the background factors that you did not control by explaining more thoroughly how these factors can have an impact on the results?

Aside from that, I have found your study interesting and well-designed.

Round 2

Reviewer 2 Report

In spite of some corrections the authors still neglect the vestibular problem in bilaterally implanted patients.

Instead of improved hearing bilateral implantation may also deteriorate quality of life of CI recipient  in the situation whey it does not bring advantage and the patient will be non user . In this situation it is just a waste of patients time unnecessary surgery and waste of expensive device. The second worst scenario is when the ballance deteriorates due to vestibular bilateral loss. These facts shoud also be clear in  the papers about bilateral implantations. Writing about advantages only is unfair.

The problem of vestibular areflexia due to bilateral implantatnion is generally underestemated in a literaturÄ™. I have seen recently the patient with this problem huge balance deficit and this complication dramatically deteriorated his quality of life. The problem is irreparable here . The patient was explained in CI center that this should be compensated in the future. It will never does because the labyrinths are not functioning.

Children’s parents  with this problem are usually explained that they had associated disorders. When vestibular test are done and it is clear that they have vestibular areflexia.

Not many centers accept  it , but it should be well knonw   in  CI centers that vestibular system is very important and should be checked if  possible before second implant especially in adult population when diagosis is easy. It does not cost much but may prevent serious problems.

The authors made some subtle changes in the paper, but generally they neglected the vestibular problem. It should be clearly stated that vestibular system is important  in bilateral implantation and this examination  - vestibular testing should have been performed in this gropu of patients , but  have not been performed here. In this patient group fortunately bilateral vestibular areflexia did not occur. This paper have not menioned about  this important preimplantation dianostic procedure.

They should explain the problem not by just mentioning about possible subjective vertigo but they should state that the problem of bilateral vestibular areflexia due to bilateral implantation may be  severe and irreparable , and may  influence motor and balance function of bilateral CI recipient .

This statement has to be changed !!!!!!!!

Bilateral sequential implantation in adults without caloric should never be performed. In my opinion and either it should be clearly staed , or the paper should not be published.

At this form I do not accept this paper for publication !!!

Moreover the authors  statement that electrical stimulation may improve  vestibular function in cochlear implant patients cannot be submitted in this place because it could be understood that the patient with ballance problem should have cochlear implant for vestibular function improvement. 

The well know paper of canadian group about this phenomenon was published as an unique observation of few unilateral CI recipients and should be removed from this place since it changes and ammaliorates the idea of vestibular lesion warning before bilateral CI implantation. The same Canadian group have described and empasised the significance of total bilateral vestibular loss (TBVL), which they previously identified as an important risk factor associated with multiple head trauma and increased likelihood of cochlear implant (CI) dysfunction. Many children had cochlear implant disfunction due to trauma caused by huge balance problems caused by vestibular areflexia – this was clear in their observation.

Wolter NE1. Gordon KA, Papsin BC, et al. Vestibular and balance impairment contributes to cochlear implant failure in children.OtolNeurotol2015;36:1029–34

Cochlear implant centers and community should be aware that there are there are single reports of improved vestibular functionafter CI activation , and single reports showing nostatistically significant influence of CI on vestibular func-tion (13). However, it is well established that CI can significantly impair vestibular system function and the worst form of it  is bilateral vestibular loss.

I will look very much forward for authors response.